# The Pathogen-Induced MATE Gene *TaPIMA1* Is Required for Defense Responses to *Rhizoctonia cerealis* in Wheat

**DOI:** 10.3390/ijms23063377

**Published:** 2022-03-21

**Authors:** Qiang Su, Wei Rong, Zengyan Zhang

**Affiliations:** The National Key Facility for Crop Gene Resources and Genetic Improvement, Institute of Crop Sciences, Chinese Academy of Agricultural Sciences, Beijing 100081, China; suqiangcaas@163.com (Q.S.); 18931648844sabrina@gmail.com (W.R.)

**Keywords:** wheat (*Triticum aestivum*), defense, *Rhizoctonia cerealis*, multi-antimicrobial extrusion family, *TaPIMA1*

## Abstract

The sharp eyespot, mainly caused by the soil-borne fungus *Rhizoctonia cerealis*, is a devastating disease endangering production of wheat (*Triticum aestivum*). Multi-Antimicrobial Extrusion (MATE) family genes are widely distributed in plant species, but little is known about MATE functions in wheat disease resistance. In this study, we identified *TaPIMA1*, a pathogen-induced MATE gene in wheat, from RNA-seq data. *TaPIMA1* expression was induced by *Rhizoctonia cerealis* and was higher in sharp eyespot-resistant wheat genotypes than in susceptible wheat genotypes. Molecular biology assays showed that TaPIMA1 belonged to the MATE family, and the expressed protein could distribute in the cytoplasm and plasma membrane. Virus-Induced Gene Silencing plus disease assessment indicated that knock-down of *TaPIMA1* impaired resistance of wheat to sharp eyespot and down-regulated the expression of defense genes (*Defensin*, *PR10*, *PR1.2*, and *Chitinase3*). Furthermore, *TaPIMA1* was rapidly induced by exogenous H_2_O_2_ and jasmonate (JA) treatments, which also promoted the expression of pathogenesis-related genes. These results suggested that *TaPIMA1* might positively regulate the defense against *R. cerealis* by up-regulating the expression of defense-associated genes in H_2_O_2_ and JA signal pathways. This study sheds light on the role of MATE transporter in wheat defense to *Rhizoctonia cerealis* and provides a potential gene for improving wheat resistance against sharp eyespot.

## 1. Introduction

Common wheat (*Triticum aestivum*) is an important staple crop for global human consumption [1]. Various diseases cause great yield losses of wheat worldwide [2]. Sharp eyespot is a damaging soil-borne disease of wheat in many regions of world [3]. This disease is caused by the necrotrophic fungal pathogen *Rhizoctonia cerealis*, which mainly infects plant stems and sheaths [3,4]. China is the largest epidemic region, where at least 6.67 million hectares of wheat plants have been infected with *R. cerealis* every year since 2005 [5,6]. *R. cerealis* also infects other cereal crops (such as barley, oats, and rye), and sugar beet, cotton, and potato [3,7,8]. To date, no completely resistant wheat germplasm to *R. cerealis* has been identified, and chemical control is still limited [9]. Various studies have shown that using important resistance-associated genes would generate wheat with better resistance to sharp eyespot [5,6,9,10,11,12,13,14]. Therefore, exploration of effective resistance genes is a viable strategy to breeding wheat with resistance to sharp eyespot.

To ward off invading pathogens, plants have evolved a two-tiered innate immune system: pathogen-associated molecular pattern (PAMP)-triggered immunity (PTI), and effector-triggered immunity (ETI) [15]. By using an *Arabidopsis**–Pseudomonas syringeae* pathosystem, studies revealed that PTI and ETI are initiated by distinct activation mechanisms and involve different early signaling cascades [16,17]. Both produce reactive oxygen species (ROS, such as H_2_O_2_ and O^2−^) and induce expression of distinct or certain common defense-associated genes in the late response signals [18]. In *Arabidopsis*
*thaliana*, previous studies showed that the up-regulated transcripts of some defense-associated genes by pathogen infection are consistent with their defensive function [19,20,21]. In wheat, several pathogen-induced defense-associated genes, such as *Ta**AGC1*, *TaRCR1*, *Ta**CRK3*, *TaMKK5*, *TaGATA1*, and *TaSTT3b-2B*, positively regulate the defense responses against *R. cerealis* [6,10,11,22,23,24]. These overexpressed genes can up-regulate the expression of downstream pathogenesis-related genes, such as *PR-1.2*, *PR2*, *chitinase3*, and *defensin* [10,11,22,23,24,25,26,27,28], resulting in the enhanced resistance to sharp eyespot in wheat [10,11,22,23,24].

Plants produce hormones, such as the salicylic acid (SA), jasmonates (JA), ethylene (ET), and abscisic acid (ABA), to amplify immune signaling and mediate defense responses against invasion of pathogens [29,30,31]. SA plays a positive role in resisting biotrophic and hemi-biotrophic pathogens [30,32]. JA and ET are responsible for defenses against necrotrophic pathogens and insects, while ABA is mainly involved in resisting abiotic stresses and hemi-biotrophic pathogens [30,33]. Interestingly, these hormones affect the expression of resistance/defense-associated genes to function [31,32,33,34]. For example, in wheat, *TaGATA1* responds to JA and cytokinin, and it subsequently up-regulates resistance responses to the infection of *R. cerealis* [10]. Similarly, *TaPIMP1* positively regulates defense responses to *Bipolaris sorokiniana* and drought stress through the ABA–SA signaling pathway in wheat [35]. *TaMYB29* overexpression enhances resistance to stripe rust by boosting H_2_O_2_ accumulation, *PR* genes expression, and SA signaling in wheat [36]. Therefore, various defense-associated genes perform their functions in a highly coordinated manner through complex phytohormone signaling networks in plants.

Multi-Antimicrobial Extrusion (MATE) transporters are present in bacteria, archaea, and all eukaryotic kingdoms and are membrane efflux transporters [37]. MATE genes are widely distributed in plant genomes, and their proteins are generally comprised of 400–550 amino acid residues that encompass a 12-transmembrane segment (TMS) and MatE domain [37]. In the model plant *Arabidopsis*
*thaliana*, MATE transporters have been shown to function in disease resistance [38,39,40], aluminum toxicity tolerance [41,42], detoxification of heavy metals [43,44], and the transport of secondary metabolites, e.g., anthocyanidins [45,46], flavonoids [47], and hormones [48,49]. Interestingly, MATE transporter, ENHANCED DISEASE SUSCEPTIBILITY 5 (EDS5), is involved in SA synthesis and is required for defense against the yellow strain of *Cucumber mosaic virus* [CMV(Y)] and *Pseudomonas syringae* pathogens in *Arabidopsis* [38,49,50,51]. Moreover, ectopic expression of *AtDTX18*, a MATE transporter controlling the extracellular accumulation of coumaroylagmatine, improves resistance of transgenic potato to *Phytophthora infestans* and *Botrytis cinerea* [40]. Conversely, *ADS1* negatively regulates *Arabidopsis* against *P. syringeae* (*Pst*DC3000) by down-regulating the accumulation of SA and the expression of *PR1* homolog genes [39]. Similarly, ectopic expression of *OsMATE1* and *OsMATE2* in *Arabidopsis* negatively regulates resistance to *Pst*DC3000 and affects plant growth and development [52]. In a previous study, RNA-seq data showed that in wheat, a MATE gene with sequence number *TraesCS2B01G296000* was upregulated and might be associated with defense to *Fusarium graminearum* [53]. However, the functional roles of MATEs in defense of *R. cerealis* remain unknown in wheat.

In this study, we identified a pathogen-induced MATE transporter named TaPIMA1 from a set of wheat RNA-seq transcriptome data and revealed that TaPIMA1 participated in resistance responses to *R. cerealis* in wheat. The expression of *TaPIMA1* was induced by *R. cerealis*, H_2_O_2_, and JA. Virus-Induced Gene Silencing (VIGS) and disease assessment results showed that *TaPIMA1* was required for wheat resistance to sharp eyespot. *TaPIMA1* positively regulated the expression profiles of at least 4 PR genes.

## 2. Results

### 2.1. Identification of TaPIMA1 by Transcriptomic Analysis

To identify resistance-related genes to *R. cerealis* in wheat, we analyzed RNA-seq data of the sharp eyespot resistant and susceptible recombinant inbred lines (RILs, derived from Shanhongmai×Wenmai6) [11]. In previous studies, some MATE transporters, e.g., AtEDS5, AtADS1, OsMATE1, OsMATE2, were shown to play important roles in disease resistance responses in *Arabidopsis* and rice [38,39]. Herein, we focused on mining wheat MATE transporters that were up-regulated in the resistant RILs than in the susceptible RILs. As a result, a MATE gene with sequence number *TraesCS3B02G563500.1*, named *TaPIMA1*, showed much higher expression levels in resistant RILs (RIL-R) compared with susceptible RILs (RIL-S) (Figure 1A). Furthermore, qRT-PCR analysis showed that *TaPIMA1* expression was significantly induced by *R. cerealis* in the resistant wheat cultivars Shanhongmai and CI12633 from 1 to 10 days post-inoculation (dpi) (Figure 1B). Subsequently, we investigated the expression profiles of *TaPIMA1* in different wheat cultivars at 10 dpi with *R. cerealis*. As expected, the transcript level of *TaPIMA1* was significantly higher in the resistant cultivars (Shanhongmai and CI12633) than in susceptible cultivars (Yangmai16 and Wenmai6), and the highest expression level was detected in Shanhongmai (Figure 1B,C). Moreover, at the filling stage of CI12633 plants, *TaPIMA1* was highly expressed in roots and stems, where sharp eyespot mainly occurred (Figure 1D). Taken together, the above results suggested that *TaPIMA1* might function in defense responses to *R. cerealis* in wheat.

### 2.2. Sequence and Phylogenetic Analyses of TaPIMA1

The cDNA and genomic sequences of *TaPIMA1* were cloned from wheat cultivar CI12633 and determined by Sanger sequencing. Sequence alignment showed that the cDNA sequence of *TaPIMA1* displayed 100% identity with the reference sequence *TraesCS3B02G563500.1*. In addition, the genomic sequence of *TaPIMA1* contained six introns and seven exons, which was transcribed into a 1500 bp-length coding (CD) sequence (Figure 2A). The TaPIMA1 protein has 499 amino acid (aa) residues, and its molecular weight is predicted to be 53.50 kDa. Moreover, the TaPIMA1 protein includes two MatE domains (no. 50–210 aa and no. 271–434 aa, respectively) and 12 transmembrane helices (TMHs, no. 445–464 aa) (Figure 2B).

The MATE family genes are functionally diverse. To determine the structural similarity of TaPIMA1 to other MATE transporters in plants, we performed phylogenetic analysis of TaPIMA1 and 24 other MATE proteins from wheat, rice, *Hordeum vulgare*, *Triticum urartu*, *Setaria viridis*, *Brachypodium distachyon*, *Sorghum bicolor*, *Arabidopsis*, *Arachis hypogaea*, *Gossypium hirsutum*, *Medicago truncatula*, and *Vitis vinifera* (Figure 2C, Appendix A). These 24 known-function MATE proteins encompass all the reported functions of the MATE transporters, such as disease resistance [38,39,40,49,50,51,52,53,54], aluminum tolerance [41,42,55,56,57,58,59,60], iron translocation [56,61,62,63], anthocyanidin transport [47,64,65], and heavy metals detoxification [43,44,66]. As a result, the dendrogram showed that these 25 MATE proteins were mainly clustered into two clades. TaPIMA1 and most of the defense-related MATEs, including OsMATE1, OsMATE2, AtADS1, and TraesCS2B01G296000, were clustered into the group I (Figure 2C). TaPIMA1 was closely related to the anthocyanidins transporters *Arabidopsis* AtTT12 and *G. hirsutum* GhTT12 (Figure 2C). The full-length of TaPIMA1 shared 47.35% and 46.47% identities with GhTT12 and AtTT12, respectively (Appendix A). These preliminary analyses suggest that the MATE protein TaPIMA1 might be involved in defense and/or anthocyanidin transport.

### 2.3. Subcellular Localization of TaPIMA1 Protein

To investigate the subcellular location of TaPIMA1 in wheat, we performed the protein transient expression assay in wheat protoplasts. The *TaPIMA1* was introduced into the PH16318 construct that was driven by the 35S CaMV promoter. Then, the PH16318-TaPIMA1-GFP and PH16318-GFP (control) construct DNAs were introduced into wheat mesophyll protoplasts and expressed, respectively. Confocal microscopic observation showed that the TaPIMA1-GFP fusion protein distributed in the cytoplasm and in plasma membrane, while the control GFP was expressed throughout the cell (Figure 3). Therefore, the results suggested that TaPIMA1 localized in wheat cytoplasm and plasma membrane.

### 2.4. Knock-Down of TaPIMA1 Reduced Resistance to Sharp Eyespot in Wheat

To specifically knock-down *TaPIMA1*, a 216 bp cDNA fragment specific to *TaPIMA1* was sub-cloned in an antisense orientation into the multi-clone site of the γ chain of *Barley stripe mosaic virus* (BSMV), generating the recombinant *γ-TaPIMA1* (Appendix A). The BSMV *α*, *β*, *γ-TaPIMA1* and *γ-GFP* construct DNA was individually transcribed into RNA in vitro (Appendix A). Subsequently, these virus RNAs (BSMV *α*, *β*, and *γ-TaPIMA1* or *α*, *β*, and *γ-GFP**)* were mixed and inoculated into the emerged third leaves of the resistant wheat cultivar CI12633 plants to execute the BSMV-mediated VIGS (BSMV-VIGS). At 10 dpi with BSMV:TaPIMA1 and BSMV:GFP, the BSMV symptom was exhibited on the new leaves of both BSMV:GFP-infected and BSMV:TaPIMA1-infected plants (Figure 4A). Additionally, the BSMV *coat protein* (*CP*) gene was detected in BSMV-infected plants by RT-PCR (Figure 4B). Moreover, the qRT-PCR showed that the mRNA levels of *TaPIMA1* were significantly decreased in BSMV:TaPIMA1-infected CI12633 plants compared with the BSMV:GFP-infected CI12633 plants (Figure 4C). These results indicated, *TaPIMA1* was successfully knocked-down in BSMV:TaPIMA1-infected CI12633 plants.

Next, we assessed disease severity of BSMV-infected wheat plants after inoculation with *R. cerealis*. At 10 dpi with *R. cerealis*, the sharp eyespot symptoms exhibited on the leaf sheaths and stems of BSMV-infected CI12633 plants, while the lesions of BSMV:TaPIMA1-infected plants were larger than those of control plants (Figure 4D). Furthermore, the fungal biomass, measured by the transcriptional level of *R. cerealis Actin*, was significantly higher in BSMV:TaPIMA1-infected plants (more than 29.8-fold) than that in control plants (Figure 4E). At ~30 dpi with *R. cerealis*, there were more serious lesions on the stems of BSMV-infected CI12633 plants, where the lesions were significantly larger in BSMV:TaPIMA1-infected plants compared with BSMV:GFP-infected plants (Figure 4F,G). The average necrotic length and width of BSMV:TaPIMA1-infected plants were 1.70 and 0.55 cm, whereas the BSMV:GFP-infected plants were 0.98 and 0.28 cm, respectively (Figure 4G). Two batches in functional assessments indicated that the average infection types (ITs) and disease index (DIs) of TaPIMA1-silenced CI12633 plants were 2.29 /2.67 and 45.88/53.33, while those of BSMV:GFP-infected CI12633 plants were 1.25/1.27 and 25.00/25.41, respectively (Figure 4H, Appendix A). These results indicated that silencing of *TaPIMA1* significantly reduced the wheat resistance to sharp eyespot, and suggested that *TaPIMA1* is required for wheat resistance to *R. cerealis*.

### 2.5. Knock-Down of TaPIMA1 Decreased the Expression of PR Genes

Previously studies have shown that the defense-associated even *PR* genes (including *PR1.2*, *PR10*, *Chitinase3*, and *defensin*) are involved in resistance responses to *R. cerealis* infection in wheat [5,22,25,26,27]. To investigate the regulatory pathway of *TaPIMA1* in response to *R. cerealis* infection, we examined the expression profiles of several *PR* genes in *TaPIMA1*-silenced CI12633 plants. As shown in Figure 5, the transcriptional levels of *PR1.2*, *PR10*, *Chitinase3*, and *defensin* were significantly downregulated in *TaPIMA1*-silenced plants compared with the control (BSMV:GFP) plants (Figure 5A–D). These results indicated that the *TaPIMA1* positively regulated the expression of *PR* genes, resulting in enhanced resistance to *R. cerealis*.

### 2.6. TaPIMA1 and Its Regulated PR Genes Were Induced by Exogenous H_2_O_2_ and JA Stimuli

The ROS, JA and SA, play important roles in plant defense responses to pathogens [17,33,34]. Thus, we analyzed the expression profiles of *TaPIMA1* in wheat CI12633 plants treated by exogenous H_2_O_2_, JA, or SA. After H_2_O_2_ treatment, the expression level of *TaPIMA1* was dramatically elevated from 0.5 h to 12 h and peaked at 3 h (~25.64-fold over non-treatment) (Figure 6A). Upon MeJA stimulus, the transcript level of *TaPIMA1* was induced from 0.5 h to 6 h and reached a peak at 0.5 h (~2.19-fold) (Figure 6B). However, *TaPIMA1* was barely responsive to exogenous SA stimulus (Appendix A).

Further, the expression profiles of *PR*genes that are regulated by *TaPIMA1* were examined in CI12633 plants treated by H_2_O_2_ or JA. Compared with mock-treatment, the transcription levels of *PR1.2*, *PR10*, and *Chitinase3* were significantly increased after treatment with H_2_O_2_ for 0.5 h and 3 h (Figure 6C–E). Similarly, *PR1.2*, *PR10*, and *Chitinase3* were significantly up-regulated by MeJA (Figure 6F–H). These data indicated that *TaPIMA1* and its regulated *PR* genes were significantly induced by H_2_O_2_ and JA.

## 3. Discussion

Wheat provides 20% of the total daily calorie expenditure of human beings in the world, but its production is threatened by sharp eyespot [5]. Developing a resistant wheat variety with disease-resistance genes is one optimal strategy to control this disease. In this study, we provided evidence that a novel MATE gene, *TaPIMA1*, was required for resistance responses to *R. cerealis* in wheat. Here, based on the RNA-seq data and qRT-PCR analysis, we identified *TaPIMA1* that was in response to *R. cerealis*, while the expression of *TaPIMA1* was significantly higher in *R. cerealis*-resistant wheat genotypes compared with the susceptible wheat genotypes (Figure 1). Sequence analysis showed that the TaPIMA1 deduced protein includes two MatE domains and 12 TMHs, similar to those of two rice disease-resistant transporters, OsMATE1 and OsMATE2 [52]. A phylogenetic analysis further indicated that TaPIMA1 was classified into group I, including MATE disease-resistant transporters OsMATE1, OsMATE2 [52], AtADS1 [39], and TraesCS2B01G296000 [53], and anthocyanidins transporters *Arabidopsis* AtTT12 and *G. hirsutum* GhTT12 [46,47]. TaPIMA1 was closely related to the anthocyanidins transporters *Arabidopsis* AtTT12 and *G. hirsutum* GhTT12. Thus, the MATE protein TaPIMA1 might participate in the wheat defense and/or anthocyanidins transport. Generally, the MATE transporters are localized in the cytoplasm and plasma membrane to perform their transporting functions [67]. For instance, the pleiotropic anti-disease transporter EDS5 is localized in cytoplasm and exports the innate immune signal SA from chloroplast [49]. Herein, TaPIMA1 was confirmed to localize in the wheat cytoplasm and plasma membrane (Figure 3).

In *Arabidopsis*, there are 56 MATE transporters that play important roles in plant growth, development, and resistance to biotic and abiotic stresses [37,67]. However, only a few MATE transporters with disease resistance, including resistance transporter EDS5 to viral and bacterial pathogens [38,68], *P. infestans* and *B. cinerea* resistance transporter AtDTX18 [40,69], and *P. syringeae* susceptibility transporter *ADS1* [39], have been characterized. The studies on functions of MATE transporters in plant immunity/defense are limited in crops, particularly in wheat. In this study, we reported that knock-down of *TaPIMA1* significantly reduced the resistance of wheat to sharp eyespot (Figure 4). Furthermore, the *TaPIMA1* transcript level was induced by *R. cerealis* and higher in root and stem tissues where sharp eyespot initially appeared; importantly, the gene transcript is higher in resistance wheat genotypes than in susceptible wheat genotypes. These data indicated that TaPIMA1 was involved in wheat innate immunity responses to *R. cerealis*.

Some upstream immune genes substantially induce the expression of defense-related genes [23,24]. For example, heightened expression of the defense-associated genes, such as *TaSTT3b-2B*, *TaRCR1*, and *Ta**PIE1*, confer enhanced resistance to *R. cerealis* in transgenic wheat [6,11,23,24]. Conversely, silencing of these genes down-regulated the expression of defense genes and weakened resistance of wheat to sharp eyespot disease [10,24,36]. Herein, the transcriptional levels of *PR1.2*, *PR10*, *Defensin*, and *Chitinase3* were significantly decreased in *TaPIMA1*-silenced wheat plants, which is consistent with the performance of wheat to sharp eyespot (Figure 5). In response to ROS, plants will up-regulate defense genes, induce callose deposition, and perform hypersensitive cell death to resist pathogens [22,70]. JA and SA, acting as immune signal amplifiers, play important roles in plant responses to necrotrophic pathogens and biotrophic pathogens, respectively [29,30,31]. In this study, *TaPIMA1* responded rapidly to exogenous H_2_O_2_ and MeJA stimuli (Figure 6A,B). Further, the expression of *TaPIMA1*-activated defense genes (*PR10*, *PR1.2*, and *Chitinase3*) was significantly up-regulated after H_2_O_2_ and MeJA treatments (Figure 6C–H). In *Arabidopsis*, EDS5 contributes to the transport and accumulation of SA, while exogenous SA stimulation induces the expression of *EDS5* and enhances the resistance to biotrophic pathogens *P. syringae* [51]. Unlike EDS5, involved in the SA-mediated resistance pathway [38,49,51], TaPIMA1 did not response to SA but exhibited a strong inductive response to JA, similar to the MATE transporter AtDTX18 [69]. Taken together, these results suggested that H_2_O_2_ and JA might participate in the *TaPIMA1*-mediated resistance to *R. cerealis* in wheat.

## 4. Materials and Methods

### 4.1. Plant and Fungus Materials, Vectors, and Primers

The resistant wheat cultivars (CI12633 and Shanhongmai) and highly susceptible wheat cultivars (Wenmai6 and Yangmai16) were used in this study. The expression profile of *TaPIMA1* was studied in CI12633, Shanhongmai, Wenmai6, and Yangmai16. The CI12633 and Yangmai16 plants were used for BSMV-VIGS experiments. All above wheat varieties were planted in a greenhouse (14 h light/10 h dark, 15–23 °C, 90% relative humidity). The sharp eyespot pathogenic fungus *R. cerealis* strain WK207 (dominant in North China) and R0301 were used in this study.

The PH16318-GFP vector was used to express fusion protein in wheat protoplast. All BSMV vectors (α, β, γ and γ-GFP) were stored in our laboratory.

The sequences of primers are listed in Appendix A.

### 4.2. Pathogen Infection and Plant Treatments

*R. cerealis* WK207 was cultured at PDA medium for 14 d. Then the *R. cerealis* WK207 was cultured with sterilized toothpicks and were covered with *R. cerealis* WK207 for 7 d at 25 °C. The leaf sheaths of wheat plants were inoculated with toothpick fragments that were covered with well-developed mycelia. The *R. cerealis* R0301 was cultured at PDA medium for 14 d and then inoculated to sterilized wheat seeds for 7 d at 25 °C. The wheat seeds carrying *R. cerealis* R0301 were inoculated into wheat roots and watered.

The CI12633 plants were grown in a greenhouse (14 h light/10 h dark, 15–23 °C, 90% relative humidity) and sprayed with 10 mM H_2_O_2_ or 0.1 mM MeJA at the 4-leaf stage. At the same time, CI12633 plants were sprayed with 0.1% Tween-20 as control for all treatments. Then, the leaves were harvested at 0 h (mock), 0.5 h, 1 h, 3 h, 6 h, and 12 h after treatment.

### 4.3. RIL Population Construction and RNA-Seq Analysis

F_10_ RILs, derived from a cross between the sharp eyespot resistant cultivar Shanhongmai and the highly susceptible cultivar Wenmai6, were created using the single seed descent method, which were kindly provided by Professor Jizeng Jia (ICS, CAAS, China). The F_11–13_ RILs were planted in Beijing (116°33′ E, 39°96′ N; 7 October–10 June) and Nanjing (118°88′ E, 32°03′ N; 7 November–31 May) to assess disease resistance against *R. cerealis* R0301 in 2013, 2014, and 2015. According to the assessment results, we selected three resistant RILs with DIs of 27.3, 28.69, and 30.42, and three highly susceptible RILs with DIs of 56.77, 67.19, and 64.19, in F_13_ generation plants for RNA-seq analysis as previously described [11]. At the tillering stage, all resistant/susceptible RILs were inoculated with *R. cerealis* WK207. Leaf sheath samples of the inoculated parts were sampled at 0 d (mock), 4 d, and 10 d after inoculation with *R. cerealis*, while three biological replicates were analyzed.

Then, the RNAs of samples were used to deep RNA-Seq based on the HiSeq 2000/2500 platform (Illumina, CA, USA) supported by Biomarker Technologies (Beijing, China). All raw data in F_ASTQ_ format (raw reads) were processed using in-house Perl scripts. Clean data (clean reads) were obtained by discarding low-quality reads, reads containing adapters and ploy-N (N > 10%). Screening for differentially expressed genes between resistant and susceptible RILs that have a log2 ratio greater than 1.0 (false discovery rate *p* < 0.05), and the expression differences between different biological replicates were consistent [11]. The number of transcript fragments per kilobase per million mapped reads for each gene was calculated based on the length of each gene and the number of fragments mapped to that gene.

### 4.4. Cloning and Bioinformatics Analysis of TaPIMA1

The full-length open reading frame (ORF) of *TaPIMA1* was amplified by nested PCR using primers pairs (TaPIMA1-FQ-F1/R1 and TaPIMA1-FQ-F2/R2) from CI12633 plants. Then, all PCR products were cloned into T vector (Takara, Tokyo, Japan) and confirmed by sequencing. The sequence of *TaPIMA1* and its homologues were referred to NCBI (https://www.ncbi.nlm.nih.gov/ (accessed on 25 December 2021)) and Ensemble Plants database (http://plants. ensembl.org (accessed on 25 December 2021)). The Clustal Omega (https://www.ebi.ac.uk/Tools/msa/clustalo/(accessed on 25 December 2021)) web program was used to perform the multiple sequence alignment. Additionally, MEGA 11 software was used to construct the phylogenetic tree by the maximum likelihood method.

### 4.5. DNA and RNA Extraction, and qRT-PCR

Genomic DNA was extracted with CTAB. The leaves of wheat plants were harvested at 10 dpi and stored at −80 °C. Total RNA was extracted using TRIzol (Invitrogen, Carlsbad, CA, USA) according to the manufacturers operating manual. Then, ~1 μg total RNA was reverse transcribed to cDNA using FastQuant RT Kit (Tiangen, Beijing, China). qRT-PCR reactions were performed on 7500 Fast Real-Time PCR Systems (Applied Biosystems, Foster City, MA, USA) using a SYBR Premix ExTaq kit (Takara, Tokyo, Japan). Additionally, each qPCR sample was run with three biological replicates and three technical replicates. The relative genes transcription levels were calculated by the 2^−ΔΔCT^ method [71]. All primers in this study for qRT-PCR are listed in Appendix A.

### 4.6. Subcellular Localization of TaPIMA1

The coding region of *TaPIMA1* without termination codon was amplified using primers TaPIMA1-sub-F/R and sub-cloned to the PH16318-GFP vector, named PH16318-GFP-TaPIMA1. The *TaPIMA1-GFP* was driven by a CaMV 35S promoter. Wheat plants were grown in pots grown in a plant growth chamber after emergence for 7 days. The seedlings were collected to produce the wheat protoplasts referring to the protocol of Liu et al. [10]. Then, ~10 μg plasmid DNA of PH16318-GFP (control) and PH16318-GFP-TaPIMA1 were introduced into wheat protoplasts and cultured for 20 h, respectively. Finally, the incubated protoplasts were imaged by confocal laser scanning microscope LSM 700 (Zeiss, Jena, Germany).

### 4.7. BSMV-VIGS in Wheat

The function of *TaPIMA1* was investigated using a *BSMV*-VIGS in wheat as described by Holzberg [72]. Briefly, a 216 bp fragment (no. 756–955) of *TaPIMA1* ORF was cloned from CI12633 plants and then subcloned into γ vector in an antisense orientation that constructed the γ-TaPIMA1 vector. The RNA of *α*, *β*, *γ-TaPIMA1* and *γ-GFP* were transcribed in vitro by RiboMA Large Scale RNA Production System-T7 kit (Promega, Madison, WI, USA) according to the method described by Zhu [6]. All transcripts were mixed with and inoculated into the wheat plants at three-leaf stage. After 10 d, the fourth leaves of inoculated seedlings were harvested to extract the total RNA.

### 4.8. Assessment of Response in BSMV-VIGS Plants to R. cerealis

After 20 d infection with BSMV, the leaf sheaths of infected wheat plants were inoculated with toothpick fragments carrying *R. cerealis* WK207. To promote the infection of *R. cerealis*, the inoculated site was tied up with wet absorbent cotton that was sprayed with ddH_2_O every day. At 30 dpi with WK207, the infection types (ITs) and disease indexes (DIs) of CI12633 plants were evaluated as described previously [5,6]. The DI = ∑(*d_i_* × *l_i_*) × 100/(*L* × *d_i_*max). The *d_i_*, *l_i_*, and *L* represent infection type, the number of plants, and the total numbers of plants for disease assessment, respectively.

## 5. Conclusions

We identified the MATE gene *TaPIMA1* participating in wheat defense responses to *R. cerealis*. The *TaPIMA1* transcript was induced by infection of *R. cerealis* and was higher in sharp eyespot-resistant wheat genotypes than in susceptible wheat genotypes. Functional dissection revealed that TaPIMA1, acting as a positive regulator, is required for resistance to sharp eyespot and for the expression of PR genes including *PR1.2*, *PR10*, *Defensin*, and *Chitinase3* in wheat. *TaPIMA1* and its regulated PR genes are in response to exogenous H_2_O_2_ and JA stimuli, suggesting that H_2_O_2_ and JA might participate in the *TaPIMA1*-mediated resistance to *R. cerealis* in wheat. This study provides insights into role of the wheat MATE in plant innate immunity to necrotrophic fungal pathogens. *TaPIMA1* is a potential gene for improving resistance to sharp eyespot in wheat.

## Figures and Tables

**Figure 1 ijms-23-03377-f001:**
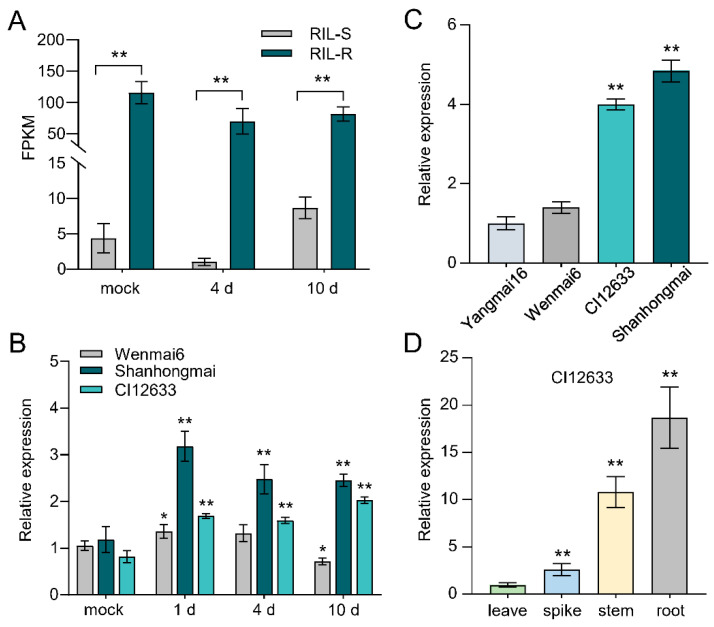
The expression profiles of *TaPIMA1* in wheat with *R. cerealis* infection. (**A**) The expression pattern of *TaPIMA1* in RNA-seq data of RILs. The sharp eyespot resistant/susceptible RILs were infected with *R. cerealis* and sampled at 4 and 10 dpi. (**B**) Transcript profiles of *TaPIMA1* in sharp eyespot-resistant cultivar CI12633 and the susceptible wheat cultivar Wenmai6 at 1, 4, and 10 dpi with *R. cerealis.* (**C**) The expression patterns of *TaPIMA1* in four wheat cultivars. All transcript data of *TaPIMA1* were compared with those in Yangmai16. (**D**) Transcriptional analyzes of *TaPIMA1* in different organs of CI12633 plants. The significant differences determined by one-way ANOVA (* *p* < 0.05, ** *p* < 0.01). Error bars indicates standard deviation.

**Figure 2 ijms-23-03377-f002:**
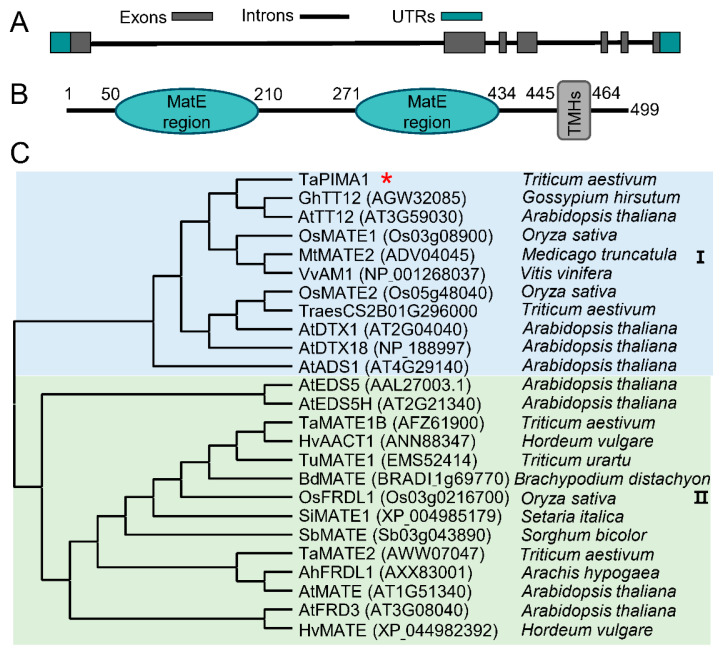
Sequence and phylogenetic analyses of TaPIMA1. (**A**) The genomic structure of *TaPIMA1* in wheat CI12633 plant. The grey frames, green frames, and lines represent exons, UTRs, and introns regions, respectively. (**B**) Schematic of TaPIMA1 protein. There are two MatE domains (no. 50–210 aa and no. 271–434 aa, respectively) and a transmembrane region (12 TMHs, no. 445–464 aa) in TaPIMA1 protein. (**C**) Phylogenetic analysis of the TaPIMA1 and other MATE proteins by the maximum likelihood method. Asterisk indicates this TaPIMA1 protein. The phylogenetic tree was constructed using MEGA 11 software. The sequences were referred to EnsemblPlants, GenBank, and Phytozome database.

**Figure 3 ijms-23-03377-f003:**
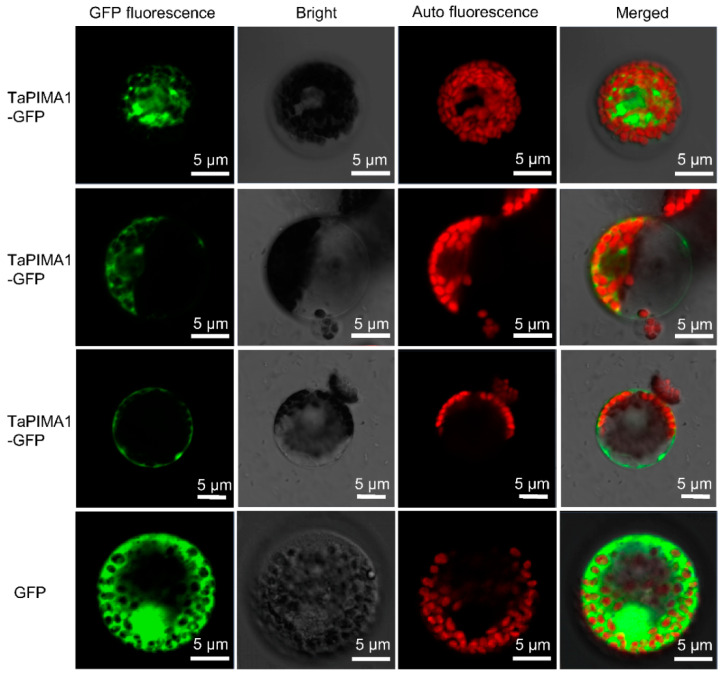
Subcellular localization of TaPIMA1 in wheat protoplasts. The GFP (control) and fused TaPIMA1-GFP were transiently expressed in wheat chloroplasts, respectively. The confocal images were taken using 488 nm wavelengths. Scale bars are each 5 μm.

**Figure 4 ijms-23-03377-f004:**
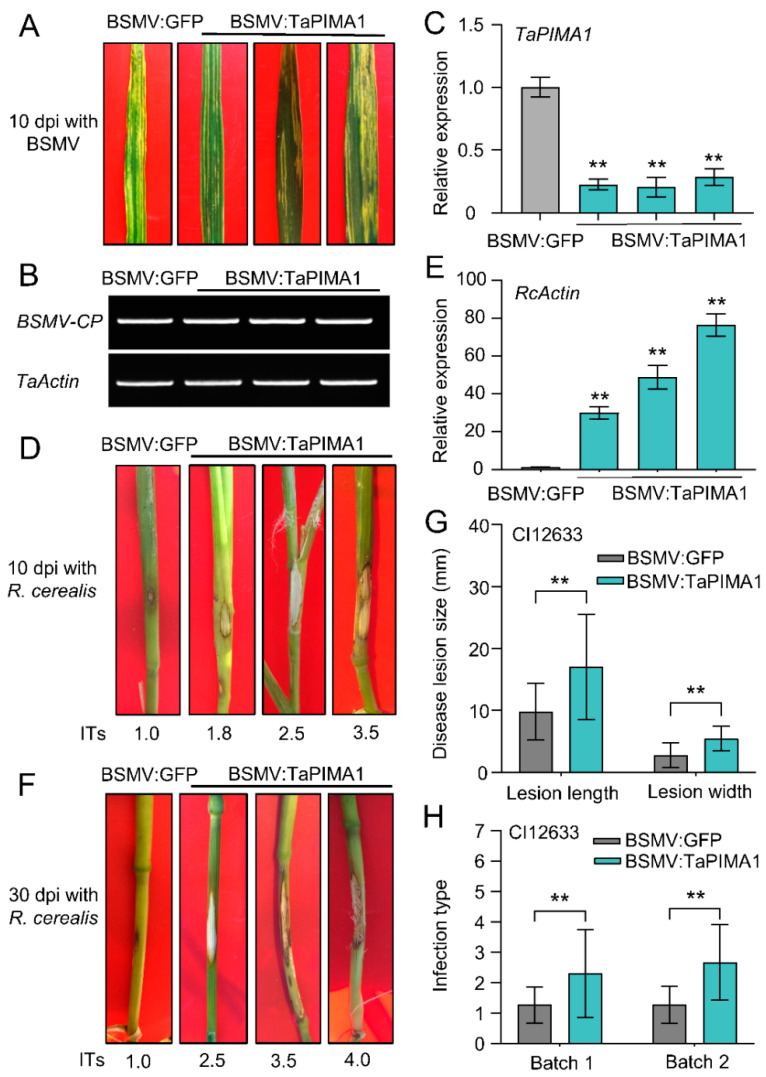
Silencing of TaPIMA1 by BSMV-VIGS reduced resistance to *R. cerealis* in CI12633 plants. (**A**) The mild chlorotic mosaic symptoms were displayed on the leaves of BSMV-infected CI12633 plants at 10 dpi. (**B**) RT-PCR analysis of BSMV *CP* in the CI12633 wheat plants. The *TaActin* (TraesCS5B02G124100.1) was set as an internal control. (**C**) Analysis of the expression of *TaPIMA1* in BSMV-infected CI12633 plants by qRT-PCR at 10 dpi. The sharp eyespot symptoms on stems of BSMV-infected CI12633 plants at 10 (**D**) and 30 (**F**) dpi with *R. cerealis*. Disease severity was indicated by infection types (ITs). (**E**) qRT-PCR analysis of the biomass of *R. cerealis* in BSMV-infected CI12633 plants. (**G**) Disease lesion size of *R. cerealis* in *TaPIMA1*-silencing and control CI12633 plants at 30 dpi. The lesion length and width represent the lesion size of sharp eyespot. (**H**) The ITs of the BSMV-infected CI12633 plants in two batches. dpi, days post inoculation. The significant differences determined by one-way ANOVA (** *p* < 0.01). Error bars indicates standard deviation.

**Figure 5 ijms-23-03377-f005:**
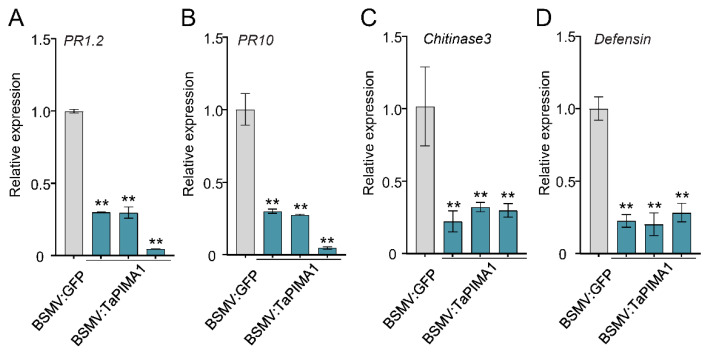
The expression levels of PR genes in *TaPIMA1*-silenced CI12633 plants at 10 dpi with *R. cerealis*. The *PR-1.2* (**A**, GenBank accession no. AJ007349), *PR10* (**B**, GenBank accession no. CA613496), *chitinase3* (**C**, GenBank accession no. LOC542780), and *defensin* (**D**, GenBank accession no. CA630387), were regulated by *TaPIMA1*. The significant differences determined by one-way ANOVA (** *p* < 0.01). *TaActin* was used as internal control. Error bars indicates standard deviation.

**Figure 6 ijms-23-03377-f006:**
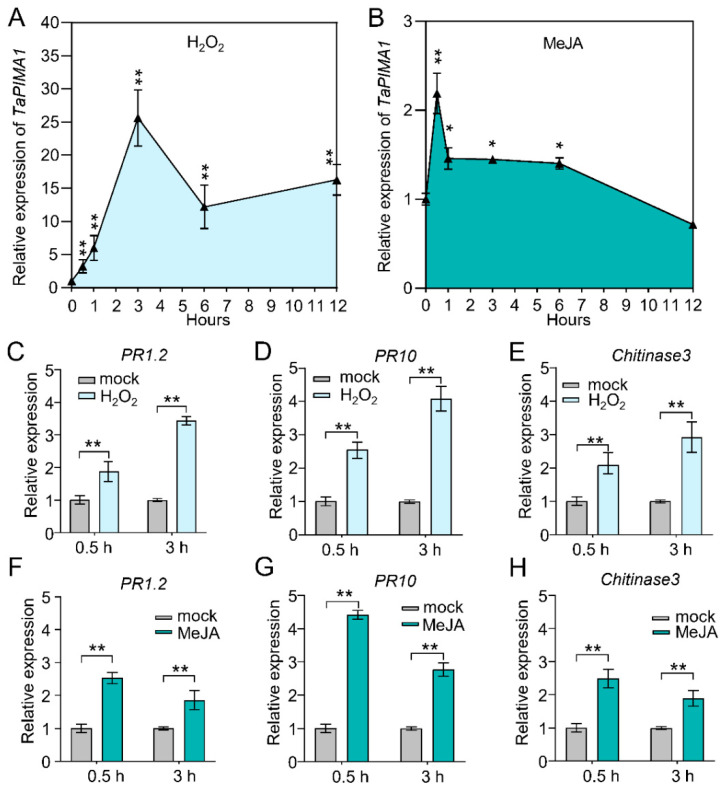
The expression profiles of *TaPIMA1* and *PR* genes in H_2_O_2_/MeJA-treated CI12633 plants. The expression profiles of *TaPIMA1* in CI12633 plants after exogenous application of H_2_O_2_ (**A**) and MeJA (**B**). Expression of *PR1.2*, *PR10*, and *Chitinase3* in H_2_O_2_ (**C**–**E**)- and MeJA (**F**–**H**)-treated CI12633 wheat plants. The CI12633 plants were sprayed with 10 mM H_2_O_2_/0.1 mM MeJA and 0.1% Tween-20 (mock) at the four-leaf stage, respectively. The significant differences were determined by Student’s *t*-test (* *p* < 0.05, ** *p* < 0.01). Error bars indicate standard deviation. *TaActin* was used as internal control.

## Data Availability

All generated and analyzed in this study can be found in the paper and its Appendix A.

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
