# Peer review of "The Pathogen-Induced MATE Gene TaPIMA1 Is Required for Defense Responses to Rhizoctonia cerealis in Wheat"

_ijms, 2022, doi:10.3390/ijms23063377_

Round 1

Reviewer 1 Report

The authors used a wheat gene belonging to the MATE family to demonstrate its involvement in establishing resistance to Rhizoctonia in wheat. They demonstrate that the candidate gene is overexpressed in R RIL-lines when compared to S, regardless of the infection with R. cerealis, and in R varieties. The authors performed knock-down experiments, using a VIGS approach, and demonstrate that the downregulation of the gene leads to increased susceptibility. The targeting of the specific gene is appropriately demonstrated. The  authors provide convincing results that few other genes known to be involved in resistance are also downregulated in silenced plants, pointing towards a role for PIMA gene as a master regulator. They also demonstrate that when exposed to elicitors of the immune system, the plants do show more expression of the PIMA gene, which can thus be a critical part of the response pathway. Based on previous literature, the authors speculate that PIMA may act as a transporter of immune signals. Overall, the study provides starting material for further analysis of the involvement of MATE genes in immune response. The model that the authors describe is very limited in scope, as data are missing, and can be eliminated as it does not add clarity.

It is not clear how the authors identified the candidate gene, besides the fact that it belongs to a promising and very numerous family of genes, and this should be made explicit. Also, it is not clear how the authors selected the MATE genes to be used in the phylogenetic tree.

I did count more than 40 inaccuracies/errors/mistyping/grammar mistakes: this should be fixed with the help of a professional editor.

Specific notes: info should be provided about the species, based on the first two letters, in the legend of figure 2; the graphic quality of figure 4 D is insufficient; materials and methods should be checked (for example, the description of the vector does not belong to "Plant and fungus materials", and an appropriate section for molecular biology materials is missing; the RILs are not described and this should be addressed; as a minor note, the authors indicate two versions of MEGA).

The manuscript needs extensive revision of the language.

Author Response

Point 1: Overall, the study provides starting material for further analysis of the involvement of MATE genes in immune response. The model that the authors describe is very limited in scope, as data are missing, and can be eliminated as it does not add clarity.

Reply 1: Thank you very much for your patience and thoughtful suggestions. According to your suggestion, we have removed Figure 7 and carefully revised the discussion part and the related conclusion.

Point 2: It is not clear how the authors identified the candidate gene, besides the fact that it belongs to a promising and very numerous family of genes, and this should be made explicit.

Reply 2: Thank you for your constructive comment. In previous studies, ~10 wheat positive immune genes were induced by R. cerealis and the induction is higher in resistant wheat genotypes than in susceptible wheat ones; importantly, those act as positive regulators and are required for wheat resistance/defense responses to the pathogen infection. It has been previously reported that MATE transporters are involved in disease resistance responses in Arabidopsis and rice. Thus, TaPIMA1 is a potential gene for improving wheat resistance to sharp eyespot. Therefore, we analyzed the differentially expressed MATE genes between sharp eyespot-resistant RILs and susceptible RILs in the RNA-seq data, and we also found TaPIMA1 expression was higher in resistant RILs than in susceptible RILs. Further qRT-PCR analysis results showed that, TaPIMA1 was induced by R. cerealis and the expression level higher in resistant wheat cultivars than in susceptible wheat cultivars. These results suggested that TaPIMA1 expression level is associated with wheat resistance. As expected, by means of VIGS and disease test, the functional analysis showed that silencing of TaPIMA1 impaired wheat resistance to sharp eyespot.

Revised portions are shown in line 115-123.

Point 3: It is not clear how the authors selected the MATE genes to be used in the phylogenetic tree.

Reply 3: The MATE transporters in plants show diverse functional roles. To initially determine the function of TaPIMA1 in wheat, we performed phylogenetic analysis of TaPIMA1 and 24 other MATE proteins from wheat, rice, Arabidopsis, Hordeum vulgare, Triticum urartu, Setaria viridis, Setaria italica, Sorghum bicolor, Arachis hypogaea, Gossypium hirsutum, Medicago truncatula and Vitis vinifera (Table 1). These 24 MATE proteins studied in other plants encompassed all the functions of the MATE transporters, such as disease resistance, iron translocation, aluminum toxicity tolerance, detoxification of heavy metals and secondary metabolites transport (e.g., anthocyanidins, flavonoids and hormones). In particular, we aligned all MATE transporters that have been reported to be associated with disease resistance in plants.

Revised portions are shown in line 155-165.

Table 1. The MATE transporters were used in phylogenetic tree analysis.

Protein name

Organism

Accession numbers

Functions

SbMATE

Sorghum bicolor

Sb03g043890

aluminum tolerance

AtADS1

Arabidopsis thaliana

AT4G29140

disease resistance

AhFRDL1

Arachis hypogaea

AXX83001

Fe-translocation/aluminum tolerance

VvAM1

Vitis vinifera

NP 001268037

anthocyanidins transport

AtFRD3

Arabidopsis thaliana

AT3G08040

iron and zinc homeostasis

AtMATE

Arabidopsis thaliana

AT1G51340

aluminum tolerance

BdMATE

Brachypodium distachyon

BRADI_1g69770

aluminum tolerance

AtDTX1

Arabidopsis thaliana

AT2G04040

detoxification of heavy metals

AtDTX18

Arabidopsis thaliana

NP_188997

detoxification

AtEDS5

Arabidopsis thaliana

NP_195614.2

disease resistance

AtEDS5h

Arabidopsis thaliana

NP_565509.4

disease resistance

GhTT12

Gossypium hirsutum

AGW32085

proanthocyanin transport

HvMATE

Hordeum vulgare

XP_044982392.1

aluminum tolerance

MtMATE2

Medicago truncatula

ADV04045.1

aluminium tolerance

OsFRDL1

Oryza sativa

Os03g08900

Fe-translocation

SiMATE1

Setaria italica

XP_004985179.1

aluminium tolerance

TaMATE1B

Triticum aestivum

AFZ61900.1

aluminum tolerance

TaMATE2

Triticum aestivum

AWW07047.1

aluminium tolerance

TuMATE1

Triticum urartu

EMS52414.1

aluminium tolerance

TaMATE2

Triticum aestivum

AWW07047.1

aluminium tolerance

HvAACT1

Hordeum vulgare

ANN88347

aluminium tolerance

OsMATE1

Oryza sativa

ABF94377.1

disease resistance

OsMATE2

Oryza sativa

AHD46250.1

disease resistance

TraesCS2B01G296000

Triticum aestivum

KAF7008897.1

disease resistance

TaPIMA1

Triticum aestivum

TraesCS3B02G563500.1

disease resistance

Point 4: Specific notes:

  • Info should be provided about the species, based on the first two letters, in the legend of figure 2.

Reply: Thank you for your careful and constructive comments. According to your suggestion, we have annotated the species corresponding to each MATE transporter in Figure 2. Revised portions are shown in line 157-159, 175.

  • The graphic quality of figure 4 D is insufficient.

Reply: We replaced the picture in Figure 4D, which has good quality. Revised portions are shown in line 232.

  • The description of the vector does not belong to "Plant and fungus materials", and an appropriate section for molecular biology materials is missing.

Reply: We revised materials and methods according to your constructive suggestion. The "Plant and fungus materials"changed to “Plant and fungus materials, vectors, and primers”. Revised portions are shown in line 377. And all vectors in this study were displayed in this part. Revised portions are shown in line 386-387.

  • The RILs are not described and this should be addressed.

Reply: We described RILs as followed: A F10 RILs derived from a cross between the sharp eyespot-resistant cultivar Shanhongmai and the highly susceptible cultivar Wenmai6, which were kindly provided by Professor Jizeng Jia (ICS, CAAS, China). Based on assessment of F11-13 RILs to defense against R. cerealis R0301, three resistant RILs (with DIs of 27.3, 28.69, and 30.42), and three highly susceptible RILs (with DIs of 56.77, 67.19, and 64.19) in F13 generation plants were used in RNA-seq analysis. Besides, we described the construction and analysis methods of RNA-seq in detail. Revised portions are shown in line 401-424.

  • As a minor note, the authors indicate two versions of MEGA.

Reply: Thank you for your thoughtful reminder. Initially we used MEGA X for the analysis, and later in the analysis process we used a more advanced version (MEGA 11). Revised portions are shown in line 181 and 432.

Point 5: I did count more than 40 inaccuracies/errors/mistyping/grammar mistakes: this should be fixed with the help of a professional editor. The manuscript needs extensive revision of the language.

Reply 5: Thank you. The language in the manuscript was revised thoroughly.

Reviewer 2 Report

  1. Provide at least two either Cq values, melting curves, or gel electrophoresis results in supplementary files
  2. Were technical replicates performed for qRT-PCR in addition to biological replication?
  3. Proofread whole manuscript for typos and grammatical errors
  4. Please provide all the information on the growing condition of the plants
  5. Add the required information about BSA in methods in spite of referring to a previous publication, as it is an important component of this manuscript

Author Response

1. Provide at least two either Cq values, melting curves, or gel electrophoresis results in supplementary files.

Response : Thank you for your careful and constructive comments. We added the raw data of qRT-PCR (Table S4) and gel electrophoresis results (Figure S2C) according to your suggestion.

2. Were technical replicates performed for qRT-PCR in addition to biological replication?

Reply 2 : Thank you for your careful and constructive comment. We conducted at least three technical repetitions for all qRT-PCR. Please refer to the Table S4 for the raw data.

3. Proofread whole manuscript for typos and grammatical errors

Reply 3: Thank you. The language in the manuscript was revised and proofread.

4. Please provide all the information on the growing condition of the plants

Reply 4: Thank you. The CI12633, Shanhongmai, Wenmai6, and Yangmai16 plants were planted in greenhouse (14 h light/10 h dark, 15~23â—¦C, 90% relative humidity). The F11-13 RILs were planted in Beijing (116°33′ E, 39°96′ N; 7 October~10 June) and Nanjing (118°88’ E, 32°03′ N; 7 November~31 May) in 2013, 2014 and 2015 to assess disease resistance against R. cerealis R0301. The resistant/susceptible RILs were planted in greenhouse (14 h light/10 h dark, 15~23â—¦C, 90% relative humidity), and all plants were inoculated with R. cerealis WK207 at tillering stage.

Revised portions are shown in line 378-383, 390-395, and 405-408.

5. Add the required information about BSA in methods in spite of referring to a previous publication, as it is an important component of this manuscript

Reply 5: Thank you for your nice advice. The information about BSA was added in line 408-414 in the revised version.

Round 2

Reviewer 2 Report

Accept as it.